# LEARNING SCALABLE CAUSAL DISCOVERY POLICIES WITH ADVERSARIAL REINFORCEMENT LEARNING

## ABSTRACT

Learning the structure of causal graphs from observational data is a fundamental but challenging problem. Existing works focus on designing search-based methods for finding optimal causal graphs. However, search-based methods have proven low-efficient since they are naturally limited by the burdensome computation of decision criteria at every step. Consequently, they can hardly scale to larger tasks. This paper proposes a novel framework called AGCORL to learn reusable causal discovery policies, which can zero-shot generalize to related tasks with much larger sizes. Specifically, AGCORL employs an Ordering Learning (OL) agent to directly infer the order of variables taken from the observational data as input. To further improve the generalizability of the OL agent, an ADversarial (AD) agent is employed to actively mine tasks where the OL agent fails to find high-quality solutions. We theoretically prove that the AD agent significantly reduces the number of required tasks to achieve generalizability of the OL agent. Extensive empirical evaluations demonstrate the superiority of our method in both runtime and solution quality over the state-of-the-art baselines.

## 1 INTRODUCTION

Discovering and understanding causal relations is a fundamental problem not only in machine learning but also in a variety of scientific disciplines such as computational biology Friedman et al. (2000); Sachs et al. (2005), epidemiology Robins et al. (2000); Vandenbroucke et al. (2016) and economics Pearl (2009); Peters et al. (2017), as well as industrial applications such as recommendations, marketing and stock Liang et al. (2016); Varian (2016); Zhang et al. (2017). A common task of interest is *causal structure learning* also known as *causal discovery* Pearl (2009); Spirtes et al. (2000); Peters et al. (2017), which requires to identify the causal relationship of variables in observational data as a Directed Acyclic Graph (DAG). Score-based methods are a major class of causal discovery techniques, which aims to find a DAG that optimizes a certain criterion:

$$\arg\min_{\mathcal{G}} \mathcal{S}(\mathcal{G}), \text{ subject to } \mathcal{G} \in \text{DAGs}, \tag{1}$$

where $\mathcal{S}(\cdot)$ is a well-defined function scoring a DAG $\mathcal{G}$ with observed data, such as Bayesian Information Criterion (BIC) score Chickering (2002). However, Problem 1 is NP-hard as the space of DAGs increases super-exponentially with the number of graph nodes Chickering (1996); Chickering et al. (2004). To search effectively, heuristic approaches like Greedy Equivalence Search (GES) add or delete edges greedily based on local heuristics which enforce the acyclicity Chickering (2002); Nandy et al. (2018). Instead of directly searching over the DAG space, Causal Additive Models (CAM) divide the structure learning into two steps: firstly search the best variable ordering greedily, and then prune the extra edges from the fully-connected DAG derived from the ordering Bühlmann et al. (2014). These methods need to compute metrics like BIC at each searching step, which makes it challenging to scale up to large tasks.

Recent works show that Reinforcement Learning (RL) Sutton & Barto (2018) has excellent potential in causal discovery tasks. RL-BIC Zhu et al. (2020) is the first RL-based casual discovery algorithm that learns to explore the DAG search space via a regularized reward function. However, such a DAG regularizer often makes the algorithms prematurely converge to suboptimal solutions. To address the issue, CORL Wang et al. (2021) avoids the acyclicity constraints by borrowing the two-stage scheme from CAM. Specifically, CORL trains an actor to output the ordering of variables in each

search epoch and tries to find better ordering following the BIC reward. Unfortunately, CORL is still a search-based method, which can hardly scale up to realistic problems with more than hundreds of variables due to the computational cost of the BIC reward Jensen & Kong (1999); Conati et al. (1997); Andreassen et al. (1991).

Inspired by the recent successes of applying RL to combinatorial optimization problems Bello et al. (2016); Khalil et al. (2017); Kool et al. (2019), we aim to train causal discovery policies that can directly infer causal structure given the observational data as input. In such a way, a well-trained policy can be reused in a class of related tasks even with a much larger number of variables. The biggest challenge of training a reusable policy is the generalizability of the policy to new tasks. To address this challenge, we propose a novel adversarial reinforcement learning framework, where an Order learning agent (OL) and an Adversarial agent (AD) are mutually trained to mine adversarial tasks that the OL agent cannot solve and thus improve the generalizability of the OL agent. Specifically, our contributions fall into the following four parts.

1. We propose an Adversarially Generalizable Causal discovery with Ordering-based Reinforcement Learning framework (AGCORL), under which we can train causal discovery policies that directly infer the causal structures from the observational data. Different from existing works where training tasks are sampled from a pre-determined task distribution, we introduce an AD agent who actively mines the adversarial tasks for the OL agent.

2. We formulate the graph generation problem of the AD agent as a Markov Decision Process (MDP) and propose a novel Ground Truth Reward (GTR) as a fast surrogate of the computationally demanding BIC score. GTR measures the difference between the discovered structures and the ground truth structures in the generated tasks.

3. Theoretically, we show the sample complexity of training the OL agent can be improved by training on adversarial tasks mined by the AD agent. And extensive experimental results on linear and nonlinear synthetic data show that AGCORL generalizes better than pretrained baselines, and can scale to much larger tasks than baselines. Furthermore, the superiority of real-world data shows the potential of our method in practice.

## 2 RELATED WORK

Most methods for structure learning from observational data belong to two classes: independence-based and score-based methods. Our method, AGCORL, is closely related to the second class. Score-based methods cast the causal discovery problem as a combinatorial optimization problem (Problem 1). To search effectively, heuristic approaches like Greedy Equivalence Search (GES) rely on local heuristics to enforce the acyclicity and add or delete edges greedily Chickering (2002); Nandy et al. (2018). Instead of directly searching over the DAG space, Causal Additive Models (CAM) divide the structure learning into two steps: firstly search the best variable ordering greedily, and then prune the extra edges from the fully-connected DAG derived from the ordering Bühlmann et al. (2014). RL-BIC Zhu et al. (2020) designed an RL agent to explore the DAG search space guided by a regularized reward function. CORL innovatively combines RL-BIC with CAM, formulating the ordering process as a Markov Decision Process (MDP) and employing an RL algorithm to search for the optimal BIC reward during testing. Inspired by CORL's formulation, our approach in AGCORL advances this concept by training a generalizable ordering policy using reinforcement learning. This trained policy is capable of inferring the order directly in test time, without the need for further search. AGCORL's key innovation, bypassing the search process during testing, significantly speeds up execution compared to CORL's method of searching for optimal order in each test instance, thus greatly enhancing efficiency.

The above heuristic and RL methods try to find the causal graph by searching. Another promising direction of research for scaling up causal discovery is continuous-optimization methods. The key that converts the discrete optimization problem into the continuous optimization problem is the differentiable DAG constraint proposed by Zheng et al. (2018) in NOTEARS. NOTEARS searches over the linear DAGs space using an augmented Lagrangian method. GOLEM Ng et al. (2020) studied the asymptotic role of the sparsity and DAG constraints in linear cases. DAGMA Bello et al. (2022) proposed a new DAG constraint via M-matrices and a log-determinant acyclicity characterization, which has better-behaved gradients and an-order-of-magnitude-faster runtime. In order to extend NOTEARS to nonlinear settings, DAG-GNN Yu et al. (2019), a graph neural network architecture

(GNN) was proposed, which can be used to learn DAGs via the maximization of evidence lower bound. By design, a DAG-GNN uses parameter sharing, which is not well suited for most DAG learning tasks. GraN-DAG Lachapelle et al. (2020) also uses NN to model the nonlinear relationship between variables but applies the acyclicity constraint at the level of neural network paths, which achieves better performance than NOTEARS and DAG-GNN.

## 3 PRELIMINARY

**Causal Graphical Models (CGM).** A CGM is defined by a joint distribution $P_X$ over $d$-dimensional random variable $\mathbf{X} = (X_1, \ldots, X_d)$ and an underlying DAG $\mathcal{G} = (d, V, E)$, where $V = \{X_1, \ldots, X_d\}$ is the set of nodes, and $E = \{(X_i, X_j)|i, j = 1, \ldots, d\}$ is the set of directed edges from $X_i$ to $X_j$. The graph structure implies a canonical factorization of the joint distribution, which is referred to as causal factorization:

$$P(X_1, \ldots, X_d) = \prod_{j=1}^{d} P(X_j \mid \mathrm{Pa}(X_j)), \tag{2}$$

where $\mathrm{Pa}(X_j)$ represents the parents of node $X_j$ in the DAG $\mathcal{G}$, i.e., $\mathrm{Pa}(X_j) := \{X_k|(k, j) \in E\}$. We assume that the observational data $\mathbf{x}_j$ is generated by the Structural Causal Model (SCM) Pearl (2009) with additive noises:

$$X_j := f_j(\mathrm{Pa}(X_j)) + \epsilon_j, j = 1, \ldots, d \tag{3}$$

where $f_j$ represents the functional relationship between $X_j$ and its parents, and $\epsilon_1, \ldots, \epsilon_d$ denote mutually independent noises associated to each node. The SCM could be of various types, including the Linear Non-Gaussian Additive noise Model Shimizu et al. (2006) and the Post Nonlinear Model Zhang & Hyvärinen (2009), based on reasonable assumptions regarding to different scenarios.

**Causal Discovery Task & BIC Score.** A causal discovery task is a tuple with two elements: $M = (W, \mathbf{D}) \in \mathcal{M}$. $W \in \{0, 1\}^{d \times d}$ is the adjacency matrix of the underlying causal graph where $W_{ij} = 1$ denotes edge $(i, j) \in E$, and $\mathbf{D} = [\mathbf{x}_1, \ldots, \mathbf{x}_d] \in \mathbb{R}^{m \times d}$ is the dataset of the nodes where $m$ is the number of samples. Given the dataset $\mathbf{D}$, the goal of causal structure learning is to find the adjacency matrix $W$ by solving Problem 1. In previous works, they usually consider BIC which is one of the most popular criterion defined as

$$\mathrm{S}_{\mathrm{BIC}}(\mathcal{G}) = \sum_{j=1}^{d} \left[ \sum_{k=1}^{m} \log p\left(x_j^k \mid \mathrm{Pa}\left(x_j^k\right); \theta_j\right) - \frac{|\theta_j|}{2} \log m \right] \tag{4}$$

where $\theta_j$ represents the parameters of the likelihood function, which can be linear or a neural network according to $f_j$. The computational cost of the BIC score heavily depends on the size of $\theta_j$.

**Ordering-based Causal Discovery.** The problem of finding a DAG can be cast as finding the order of variables Wang et al. (2021) and then prune the fully-connected DAG generated from the inferred order. Formally, let $\Omega$ be an ordered set of variables. We denote by $\Omega_{\prec X_j}$ the set of variables preceding $X_j$ in $\Omega$. CAM Bühlmann et al. (2014) searches the order greedily and CORL Wang et al. (2021) formulate the ordering process as an MDP: at step $t$, CORL agent takes a action to pick a variable $X_j$ as the $t$-th element in $\Omega$. At the end of one episode, we have $\Omega_{\prec X_j}$ for all $j \in [d]$, so we can easily establish a unique fully-connected DAG $\mathcal{G}^{\Omega}$ whose canonical factorization is $P(X_1, \ldots, X_d) = \prod_{j=1}^{d} P\left(X_j \mid \Omega_{\prec X_j}\right)$. Then, the BIC reward can be calculated by Equation 4 to guild the searching of CORL. After all searching episodes, variable selection algorithms (Bühlmann et al. (2014); Lachapelle et al. (2020); Wang et al. (2021)) will be applied to prune the optimal $\mathcal{G}^{\Omega^*}$ to get the final DAG.

## 4 ADVERSARIAL RL FRAMEWORK FOR CAUSAL DISCOVERY

The existing search-based methods fail to scale up because they have to compute the BIC score at each iteration with a computational cost of $\mathcal{O}(d^3)$, where $d$ is the number of variables. Search-based

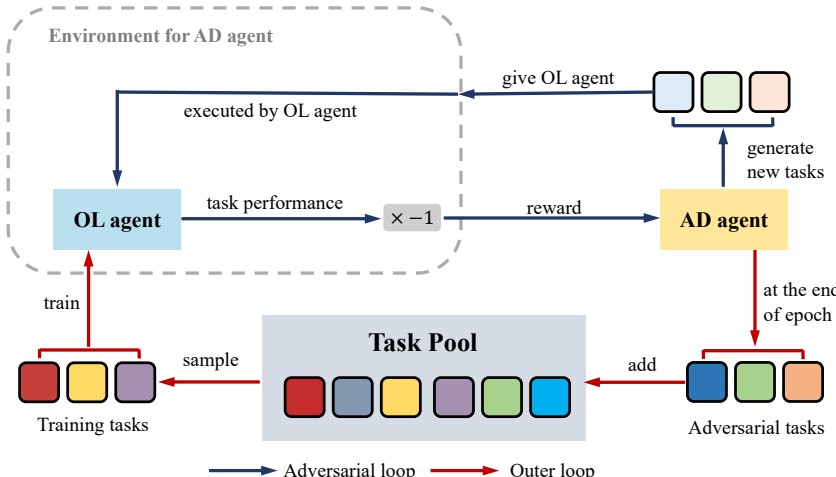

Figure 1: The overview of the AGCORL training framework

methods can hardly scale up to causal discovery tasks with a large number of variables. In this work, we aim to train a policy to directly infer the order of variables from the observational data without searching. This new approach to the causal discovery tasks has significant advantages in terms of both generalizability and scalability. Unfortunately, training such a policy is challenging because it could require massive number of training tasks. Thus, the quality of the training tasks also plays an important role. In fact, compared to easy counterparts, the tasks where the current policy fails to find high-quality solutions are more valuable. To this end, we propose Adversarially Generalizable Causal Discovery with Ordering-based Reinforcement Learning (AGCORL) framework. In the AGCORL framework, **Order Learning (OL) agent** and **ADversarial (AD) agent** are trained adversarially. OL agent is trained on a set of tasks $\mathcal{M}_{train} = \{M_1, \ldots, M_n\}$ to directly infer the order of variables given the data $\mathbf{D}$ of a causal discovery task $M$. Moreover, instead of training the policy with the tasks sampled from some pre-determined distributions, we train the other AD agent which actively mines the adversarial tasks $\mathcal{M}_{adv}$ on which the current OL agent performs poorly and adds them to the training tasks pool $\mathcal{M}_{train}$.

## 4.1 INFERRING ORDER OF VARIABLES BY OL AGENT

As is introduced in Section 3, the causal discovery task can be reduced to inferring the order of variables Bühlmann et al. (2014); Wang et al. (2021). Thus, we formulate the order search problem as an $d$-step MDP, where $d$ represents the number of variables. The basic elements of the MDP are illustrated as follows.

**Action.** The action $a_t$ at each timestep is to select a variable from the candidate variable set $V = \{X_1, \ldots, X_d\}$. Once a variable is selected, it will be removed from the candidate variable set. Then, at the end of an episode, the actions will make up an ordered set $\Omega$ consisting of all variables.

**State and Transition.** A state describes the current relationship between variables (nodes in the DAG). At the beginning of each episode, we will sample a batch of $N$ samples $[\mathbf{x}_1, \ldots, \mathbf{x}_d] \in \mathbb{R}^{N \times d}$ from dataset $\mathbf{D}$. Each variable $X_i \in V$ can be represented by an embedding $\mathbf{s}_i = \Phi(\mathbf{x}_i)$, where $\Phi$ is a standard Transformer encoder. The overall state $\mathbf{S}^t$ can be represented by a tuple $< \mathbf{S}_+^t, \mathbf{S}_-^t >$, where $\mathbf{S}_+^t$ is the set of embeddings of variables that have not been selected, and $\mathbf{S}_-^t$ is the set of embeddings of variables that have been selected. Obviously, at the initial state $\mathbf{S}_+^0 = \{\mathbf{s}_i | i = [d]\}$ contains all node embeddings and $\mathbf{S}_-^0 = \varnothing$. At the end of episode, $\mathbf{S}_+^T = \varnothing$ and $\mathbf{S}_-^T = \{\mathbf{s}_i | i = [d]\}$. Fig. 5 in Appendix illustrates how the policy network maps a state $\mathbf{S}^t$ to an action $a_t$.

**Reward.** As aforementioned, the computational cost of the BIC score prohibits the existing methods from scaling up to large problems. In fact, the computation requires performing linear regression,

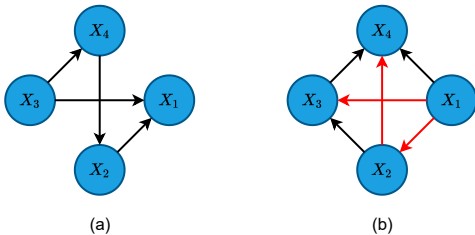

Figure 2: An example of the computation of GTR. (a) The ground truth DAG with order $\Omega^* = (X_3, X_4, X_2, X_1)$, (b) The fully-connected DAG generated from $\Omega = (X_1, X_2, X_3, X_4)$, the red arrows are reversed edges compare with (a). The final reward for $\Omega$ is $-3/4$, where 3 is the number of reversed edges and 4 is the total number of edges in (a). Note that reward will be 0 when $\Omega^* = \Omega$.

neural network training, or Gaussian process regression in every training step. On the other hand, since the training tasks are generated by the AD agent in our scenario, we have access to the corresponding ground truth DAGs during training. Therefore, we propose a fast Ground Truth Reward (GTR) to evaluate the discrepancy between the resulting DAG and the ground truth DAG. The goal of the OL agent is to minimize such a discrepancy by maximizing the GTR.

**Ground Truth Reward.** The reward aims to evaluate the order $\Omega$ by comparing its corresponding DAG $\mathcal{G}^\Omega$ with the ground truth DAG $\mathcal{G}^*$. However, many orders could correspond to the same DAG since exchanging two irrelevant variables in an order does not affect the resulting DAG. In other words, $\mathcal{G}^\Omega$ only inherits a partial order from the full ordered set $\Omega$. Therefore, the principle of designing reward should capture the differences of partial orders between $\mathcal{G}^\Omega$ and $\mathcal{G}^*$.

Suppose two nodes in $\Omega$ satisfy partial order $X_i \prec X_j$. Then, at least one path from $X_j$ to $X_i$ must be in the corresponding DAG $\mathcal{G}^\Omega$. If we reverse the partial order of the two nodes to $X_j \prec X_i$, at least one edge must be reversed in $\mathcal{G}^\Omega$. Based on this observation, we design the ground truth reward function by punishing the reversed edges comparing $\mathcal{G}^\Omega$ with $\mathcal{G}^*$. We denote by $e_\Omega^{rev}$ the number of reversed edges comparing $\mathcal{G}^\Omega$ with $\mathcal{G}^*$. In addition, since we will train multiple tasks with different numbers of variables, it is necessary to balance the rewards of different tasks. Finally, we define the episodic Ground Truth Reward of task $M$ as $\mathcal{R}_{OL}(M, \Omega) = -e_\Omega^{rev}/h$, where $\Omega$ is the final order outputted by the policy and $h$ is the total number of edges in $\mathcal{G}^*$. The negative sign indicates the punishment, as the goal of the OL agent is to maximize the reward. Fig. 2 shows an example of computing the GTR. Note that the computation of GTR requires only counting the edges and, therefore much more efficient than the computation of the BIC score.

**OL agent policy and training.** Since the policy of the OL agent sequentially selects variables at each time step, we choose the Pointer Net Vinyals et al. (2015) as the backbone of our policy network $\pi_\phi$. Fig. 5 in Appendix shows the details of the network architecture. We adopt the actor-critic method Konda & Tsitsiklis (1999) to train the OL agent, where an additional critic network $V_\psi$ is introduced to estimate the baseline value of states.

To improve the generalizability of the OL agent, we will iteratively train it over a set of tasks $\mathcal{M}_{train}$. For each task $M \sim \mathcal{M}_{train}$, the policy gradient for the actor is shown in Equation 5. Note that the reward $\mathcal{R}_{OL}(M, \Omega)$ can only be computed at the end of an episode when all variables have been selected, therefore our critic $V_\psi$ is only used to estimate the value of the initial state.

$$\nabla J(\phi) = \mathbb{E}_{\mathbf{S}^0 \sim \mathbf{D}_M} \left[ \left( \mathcal{R}_{OL}(M, \Omega) - V_\psi(\mathbf{S}^0) \right) \sum_{t=0}^{T} \nabla_\phi \log \pi_\phi \left( a^t \mid \mathbf{S}^t \right) \right] \quad (5)$$

The critic $V_\psi$ will be episodically updated by minimizing the following Mean Square Error (MSE).

$$L(\psi) = \mathbb{E}_{\mathbf{S}^0 \sim \mathbf{D}_M} \left[ \text{MSE}(\mathcal{R}_{OL}(M, \Omega), V_\psi(\mathbf{S}^0)) \right]. \quad (6)$$

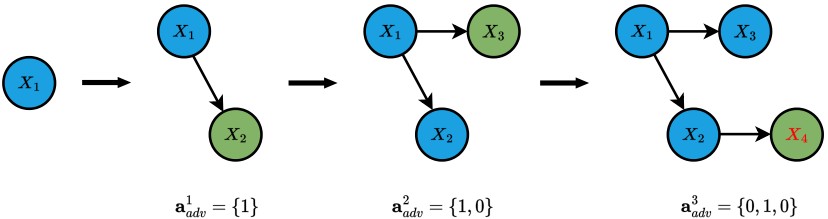

Figure 3: An example of adversarial DAG generation: the blue nodes are the nodes generated in the previous timesteps and the green node is the newly generated node. At each timestep $t$, the AD agent output $\mathbf{a}^t_{adv}$ to determine the parents of the green node. For example, $\mathbf{a}^3_{adv} = \{0, 1, 0\}$ means only $X_2$ is the parent of $X_4$. Then we will generate data of $X_4$ following Equation 3: $X_j := f_4((X_2)) + \epsilon_4$, where $f_4$ and $\epsilon_4$ are sampled from the SCM distribution and noise distribution.

## 4.2 GRAPH GENERATION WITH ADVERSARIAL AGENT

To improve the generalizability of the OL agent, we need to actively mine causal discovery tasks where the OL agent fails. We also formulate this causal discovery task generation process as an MDP. To generate a causal discovery task, one needs to determine its graph size, graph structure, type of Structural Causal Model (SCM) and the observational data. In real-world scenarios, the ground truth of SCM type is usually unknown, the SCM type used to model causal relationship is usually determined under some reasonable assumptions, such as Linear Non-Gaussian Additive noise Model (LiNGAM) and Post Nonlinear Model (PNL)Shimizu et al. (2006); Zhang & Hyvärinen (2009). In addition, the policy trained on small tasks can generalize to large ones, we keep the sizes of the generated tasks the same with that in the original datasets. Hence, the graph generation problem of our AD agent is reduced to specifying the DAG structure and the observational data associated with the nodes in the DAG. In order to reduce the action space at each step, our AD agent is designed to generate nodes one by one. We formulate the sequential decision-making process as an MDP as follows.

**State.** The state of the AD agent describes the set of variables generated so far. We denote by $\mathbf{S}^t_{adv} = \{\mathbf{s}_0, \ldots, \mathbf{s}_t\}$ the state of the AD agent at time $t$, where $\mathbf{s}_t = \Phi_{adv}(\mathbf{x}_t)$ is the embedding of the variable $X_t$ and $\mathbf{x}_t \in \mathbb{R}^N$ is the batch of data associated with $X_t$ generated at time $t$.

**Action and Transition.** An action $\mathbf{a}^t_{adv}$ is a length-$t$ binary vector sampled from a $t$-dimensional *Bernoulli* distribution, whose parameters are determined by the policy of the AD agent. The action specifies how a newly generated node $X_t$ is added to the current adversarial DAG $\mathcal{G}^t_{adv}$. Fig. 3 shows an example of constructing the adversarial DAG. Once the SCM of Equation 3 is specified, we will have a function $\mathcal{F}$ that maps the data generated so far $\{\mathbf{x}_0, ..., \mathbf{x}_t\}$ and the current $\mathcal{G}^t_{adv}$ to $\mathbf{x}_{t+1}$. At the end of the episode, we will have an adversarial task $M_{adv} = \{\mathcal{G}^T_{adv}, \{\mathbf{x}_0, \ldots, \mathbf{x}_T\}\}$.

**Reward.** The AD agent aims to find tasks that the OL agent fails to solve, so the reward for the AD agent is based on the performance of the OL agent on the generated task $\mathcal{M}_{adv}$. Therefore, we define the reward for the AD agent as $\mathcal{R}_{AD}(\mathcal{M}_{adv}, \Omega) = -\mathcal{R}_{OL}(\mathcal{M}_{adv}, \Omega)$, where $\Omega$ is the ordered set of variables inferred by the OL agent.

**AD agent policy and training.** The policy of the AD agent maps the current state to the parameters of the *Bernoulli* distribution used at the next time step. Fig. 6 in Appendix shows the architecture of the policy network of the AD agent. We also adopt the actor-critic framework Konda & Tsitsiklis (1999) to train the AD agent and reuse the critic $V_\psi$ of OL agent. As the input of $V_\psi$ should be the embedding of the full set of nodes, so here the baseline value $V_\psi(\mathbf{S}^T_{adv})$ is estimated by the terminal state $\mathbf{S}^T_{adv}$. The policy gradient for the AD agent actor $\pi_\theta$ is written as follows.

$$\nabla J(\theta) = \mathbb{E}_{\mathbf{S}^0_{adv}} \left[ (V_\psi(\mathbf{S}^T_{adv}) - \mathcal{R}_{OL}(\mathcal{M}_{adv}, \Omega)) \sum_{t=0}^{T} \nabla_\theta \log \pi_\theta \left(\mathbf{a}^t_{adv} \mid \mathbf{S}^t_{adv}\right) \right] \quad (7)$$

### 4.3 ADVERSARIAL TRAINING AND DEPLOYMENT

In this section, we introduce how to jointly train the OL agent and AD agent in our proposed adversarial training framework. All training tasks $M_{adv}$ mined by the AD agent will be stored in a set $\mathcal{M}_{train}$. The adversarial training framework can be viewed as a zero-sum game. In each training epoch, the OL agent and the AD agent are trained in turn to maximize their own rewards and minimize opponents' rewards. In each adversarial training epoch, the OL agent samples tasks from the tasks pool and learns to infer the correct order guided by GTR; then the AD agent is trained to find the tasks where the OL agent performs unsatisfactorily by minimize the performance of the OL agent on its generating tasks measured by GTR; finally the generated tasks will be added to task pool which will be learned by the OL agent in the following epochs. Please also refer to Algorithm 3 in Appendix C for details

**Deployment.** After adversarial training, the OL agent is supposed to zero-shot transfer to target tasks. However, the agent cannot take all data as input because of the data-sampling state space design. To get better performance from the probabilistic policy, we sample a batch of initial states in parallel and get a batch of ordered sets. Then we rank them by their BIC scores and select the best actions. Finally, we prune the fully-connected graph generated from the best order to obtain the final DAG. Alg. 4 in Appendix shows the details.

## 5 EXPERIMENT

In this section, we conduct experiments to verify the generalizability of our methods to tasks with different sizes, noise types, and function types and compare our method with baselines in terms of performance and scalability on synthetic linear and nonlinear tasks as well as real data sets.

**Baselines.** The baselines include random policy, the heuristic ordering-based searching approaches CAM Bühlmann et al. (2014) and CORL Wang et al. (2021), the gradient-based methods NOTEARS Zheng et al. (2018), DAG-GNN Yu et al. (2019) and GraN-DAG Lachapelle et al. (2020), and CORL-P which is CORL pretrained with presampled tasks. We use the code from the causal discovery toolbox Zhang et al. (2021).

**Data generation.** We generate testing synthetic data sets which vary along five dimensions: level of edge sparsity, graph type, number of nodes, causal functions, and sample size. We sample 10 data sets with 500 samples for each task: a ground truth DAG $G$ is firstly drawn randomly from either the *Erdős–Rényi* (ER) or *scale-free* (SF) graph model(5 from the ER graph model and the other 5 from the SF graph model) and the data are then generated according to different given Structural Equation Models (SEMs) model $X_j := f_j(\mathrm{Pa}(X_j)) + \epsilon_j, j = 1, \ldots, d$.

**Metrics.** We consider two common metrics to evaluate the performance: True Positive Rate (TPR) and Structural Hamming Distance (SHD). The former indicates the probability of finding the right edges, which is the higher, the better. The latter counts the total number of missing, false positive, or reversed edges, which is the smaller, the better.

**Pruning.** We adopt the same variable selection methods for edge pruning as CORL. For linear tasks, we apply linear regression to the obtained fully-connected DAG and then use a threshold to prune edges with small weights, as similarly used by Zheng et al. (2018). For the non-linear tasks, we adopt the CAM pruning Bühlmann et al. (2014) used by Lachapelle et al. (2020). For each variable $X_j$, one can fit a generalized additive model against the current parents of $X_j$ and then apply significance testing of covariates, declaring significance if the reported $p$-values are no greater than 0.001. Other variable selection methods can also be considered, such as sparse candidate Teyssier & Koller (2005) and group Lasso Schmidt et al. (2007).

### 5.1 LINEAR MODELS WITH GAUSSIAN NOISE

We further evaluate the proposed methods on *linear-Gaussian* (LG) tasks with equal variance Gaussian noise. We set $h \in \{2, 5\}$ and $d \in \{50, 100, 150, 200\}$ to obtain the ER and SF graphs with different

Table 1: Empirical results for DAGs of 50 and 100 nodes with LG data

| Method | | | Random | NOTEARS | CORL | CORL-P | AGCORL |
|---|---|---|---|---|---|---|---|
| 50-node | 2-edge | TPR | $0.37_{\pm 0.03}$ | $0.91_{\pm 0.07}$ | $0.92_{\pm 0.04}$ | $0.92_{\pm 0.03}$ | $\mathbf{0.94}_{\pm 0.03}$ |
| | | SHD | $161.1_{\pm 21.6}$ | $21.1_{\pm 18.9}$ | $21.4_{\pm 7.4}$ | $33.6_{\pm 11.6}$ | $\mathbf{16.1}_{\pm 6.5}$ |
| | 5-edge | TPR | $0.42_{\pm 0.02}$ | $0.70_{\pm 0.17}$ | $0.89_{\pm 0.09}$ | $0.87_{\pm 0.12}$ | $\mathbf{0.95}_{\pm 0.04}$ |
| | | SHD | $351.1_{\pm 24.3}$ | $130.8_{\pm 42.5}$ | $101.1_{\pm 17.3}$ | $172.3_{\pm 33.5}$ | $\mathbf{80.9}_{\pm 15.7}$ |
| | t | | - | 12m | $0.8h$ | **4.7s** | **4.8s** |
| 100-node | 2-edge | TPR | $0.39_{\pm 0.04}$ | $0.83_{\pm 0.01}$ | $0.91_{\pm 0.01}$ | $0.90_{\pm 0.02}$ | $\mathbf{0.93}_{\pm 0.01}$ |
| | | SHD | $394.6_{\pm 27.8}$ | $85.3_{\pm 50.0}$ | $87.9_{\pm 14.6}$ | $118.2_{\pm 21.6}$ | $\mathbf{79.3}_{\pm 11.3}$ |
| | 5-edge | TPR | $0.41_{\pm 0.04}$ | $0.64_{\pm 0.20}$ | $0.90_{\pm 0.02}$ | $0.88_{\pm 0.02}$ | $\mathbf{0.94}_{\pm 0.02}$ |
| | | SHD | $940.0_{\pm 28.5}$ | $\mathbf{303.5}_{\pm 128.6}$ | $437.3_{\pm 68.5}$ | $504.3_{\pm 89.2}$ | $360_{\pm 37.4}$ |
| | t | | - | 1h | 12h | **19.8s** | **19.2s** |

levels of edge sparsity and different numbers of nodes. Then we generate 500 samples for each task following the linear SEM: $\mathbf{X} = W^T \mathbf{X} + \epsilon$, where $W \in \mathbb{R}^{d \times d}$ denotes the weighted adjacency matrix obtained by assigning edge weights independently sampled from a uniform distribution $Unif([-2, -0.5] \cup [0.5, 2])$. Here we present the evaluation result of the proposed method and baselines on LG tasks with 50- and 100-node tasks in Table 1.

In this experiment, CORL is trained from scratch in each task for 2000 episodes. AGCORL is trained on 20-node tasks for 10 epochs. At the end of each epoch, the AD agent generates 10 adversarial tasks, which will be added to the training task pool. So the total number of training tasks is 100. CORL-P is trained for the same total of 40000 iterations as AGCORL on 200 uniformly sampled 20-node tasks. Across all settings, AGCORL is the best-performing method in terms of both TPR and SHD. For scalability, the running time of CORL is the longest due to its *interactive search by training* manner. AGCORL, which is trained on 100 actively mined tasks, is better than CORL-P, which is trained on 200 pre-sampled tasks, which shows the importance of adversarial training. We also present AGCORL's performance on larger tasks in Fig. 7 in the Appendix, the SHD increases as the number of edges increases, but the TPR only decreases a little even on 200-node tasks, which shows that our method can generalize to very large tasks.

To further illustrate the effect of adversarial training, we present the joint training curve of AGCORL on LG tasks in Fig. 4. The periodical downward spikes illustrate the adversarial training. As the amplitude of the spikes becomes smaller, the generalizability of the OL agent becomes better, and thus the testing performance becomes better too.

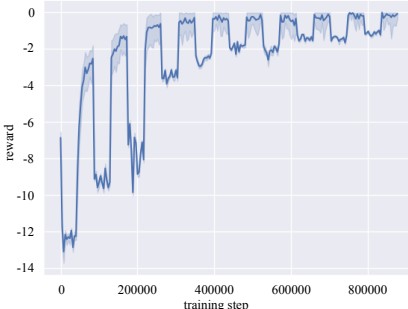
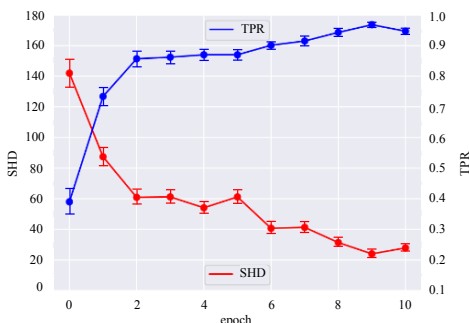

Figure 4: The left figure is training curve of AGCORL on 20-node LG tasks for 10 epochs and the right figure is the evaluation on 30-node-5-edge tasks at each epoch.

Table 2: Empirical results for DAGs of 10 and 30 nodes with GP data

| | Method | | CAM | GraN-DAG | CORL | CORL-P | AGCORL |
|---|---|---|---|---|---|---|---|
| 10-node | 1-edge | TPR | $\mathbf{0.75}_{\pm0.06}$ | $0.59_{\pm0.12}$ | $\mathbf{0.74}_{\pm0.03}$ | $0.64_{\pm0.08}$ | $\mathbf{0.74}_{\pm0.04}$ |
| | | SHD | $\mathbf{2.3}_{\pm1.1}$ | $5.2_{\pm3.3}$ | $\mathbf{2.5}_{\pm1.1}$ | $3.3_{\pm1.4}$ | $\mathbf{2.5}_{\pm1.2}$ |
| | 4-edge | TPR | $0.40_{\pm0.05}$ | $\mathbf{0.64}_{\pm0.11}$ | $0.32_{\pm0.12}$ | $0.32_{\pm0.14}$ | $0.36_{\pm0.07}$ |
| | | SHD | $18.2_{\pm3.7}$ | $25.3_{\pm4.8}$ | $20.0_{\pm3.4}$ | $21.4_{\pm3.8}$ | $\mathbf{13.6}_{\pm2.9}$ |
| | t | | 63s | 17m | 11m | **49s** | **51s** |
| 30-node | 1-edge | TPR | $\mathbf{0.73}_{\pm0.08}$ | $0.35_{\pm0.04}$ | $0.51_{\pm0.09}$ | $0.57_{\pm0.15}$ | $\mathbf{0.72}_{\pm0.06}$ |
| | | SHD | $\mathbf{11.1}_{\pm2.9}$ | $20.1_{\pm5.7}$ | $16.2_{\pm4.1}$ | $13.8_{\pm3.7}$ | $\mathbf{11.8}_{\pm2.6}$ |
| | 4-edge | TPR | $0.24_{\pm0.04}$ | $\mathbf{0.31}_{\pm0.03}$ | $0.19_{\pm0.04}$ | $0.20_{\pm0.05}$ | $\mathbf{0.21}_{\pm0.04}$ |
| | | SHD | $87.0_{\pm19.8}$ | $97.4_{\pm11.5}$ | $90.1_{\pm20.3}$ | $85.2_{\pm16.5}$ | $\mathbf{81.0}_{\pm10.7}$ |
| | t | | 53m | 30m | 12h | **11m** | **11m** |

## 5.2 Non-Linear Model with Gaussian Process

In this set of experiments, we consider a causal relationship with $f_i$ being a function sampled from the Gaussian Process (GP) with radial basis function kernel of bandwidth one. The additive noise follows standard Gaussian distribution. The GP data sets with $h \in \{1, 4\}$ and $d \in \{10, 30, 80, 100\}$ are generated following $X_j = f_j ( \mathrm{Pa}(X_j)) + \epsilon_j$, where the function $f_j$ is a function sampled from a GP with radial basis function kernel of bandwidth one and $\epsilon_j$ follows standard Gaussian distribution.

Presented in Table 2, AGCORL performs as well as CAM, but the deployment time of AGCORL is much less than CAM when the task is large. GraN-DAG gets the highest TPR in denser tasks, but the SHD is poor because it produces more edges than other methods. Besides, CORL is better than CORL-P on small tasks, but CORL-P performs better than CORL on 30-node tasks because CORL cannot converge in 2000 episodes on 30-node tasks. Like the result in LG tasks, the deployment time of CORL-P is close to AGCORL, but the performance is poor because of a lack of generalizability. Fig. 8 in Appendix shows the performance on large GP tasks, which is much more difficult than the linear case.

## 5.3 Real-world Data

We test our agent trained in Section 5.2 on a real-world data sets: Sachs et al. (2005) with 11-node and 17-edge true graph, which is widely used for research on graphical models. The expression levels of protein and phospholipid in the data set can be used to discover the implicit protein signal network. The observational data set has m = 853 samples and is used to discover the causal structure. In this experiment, AGCORL and CORL achieve the best SHD 11, which shows our AGCORL can successfully generalize to real-world data. CAM, GraN-DAG, DAG-GNN and NOTEARS achieve SHDs 12, 13, 16, and 19 respectively. However, the running times of AGCORL and CORL are 56s and 12m, which shows the superiority of AGCORL in scalability.

## 6 Conclusion

In this paper, we propose AGCORL, an adversarial training framework for training generalizable and scalable causal discovery policies. Compared to existing search-based methods, our causal discovery policies directly infer the causal graphs from the observational data, thus significantly reducing the computational cost. AGCORL employs an OL agent to infer the causal graph from data, and an AD agent to actively mine adversarial tasks where the OL fails. To further accelerate training, we design an efficient GTR function to evaluate the quality of inferred causal graphs, which provides reward signals for both agents. Our experiments show the advantages of the AGCORL framework, in terms of both solution quality and scalability. We believe that our method is particularly suitable for handling specific domains with a large number of similar causal discovery tasks. For future works, we plan to design more efficient representations of nodes on the DAG, in order to further reduce the number of tasks during training and improve data efficiency.

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

## A THEORY

### A.1 GENERALIZABLE RL AND ADVERSARIAL ROBUSTNESS

As discussed earlier, learning generalizable RL-based causal discovery policies can eliminate the need for online learning, which is the key to scaling up. However, the question of 'given the noise level of the AD agent, how many tasks/samples does one need to achieve sufficiently good performance?' should be answered. The key is to measure the performance of value function of OL agent under noise influence on AD agent. Theorem 1 gives a lower bound on the number of generated tasks required to achieve generalizability.

Let $g : \mathcal{M} \to \{0, 1\}$ be a function distinguishing whether a task $M$ can be correctly ordered by the OL agent, where $\mathcal{M}$ is the set of all tasks and $\mathcal{M}_{train}, \mathcal{M}_{adv} \subset \mathcal{M}$. $g(M) = 1$ if the OL agent can correctly order variables in $M$, otherwise $g(M) = 0$. After the OL training in each epoch, the OL agent is trained to succeed in $\mathcal{M}_{train}$. Let $supp(g)$ be the support of $g$.[1] Thus $\mathcal{M}_{train}$ is a subset of $supp(g)$, i.e. $\mathcal{M}_{train} \subset supp(g)$. Suppose that the AD agent generates task $M \in supp(g)$ with the probability $1 - \zeta$ and generates wrong tasks (i.e., $M_{adv} \notin supp(g)$) with probability $\zeta$, where $\zeta \in \{0, 0.5\}$ is the noise level of AD agent. If $\zeta = 0$, the AD agent will always generate the true tasks. The generate is fully random when $\zeta = 0.5$, in this case, the sample from AD agent can not provide sufficient information to OL agent.

**Theorem 1 (Imperfection of generation)** *Suppose that $\epsilon, \delta \in (0, 1)$, $m$ is the number of samples from each task, then we need to have $\frac{2 \log \frac{2|\Psi|}{\delta}}{2m\epsilon^2(1-2\zeta)^2}$ tasks on nodes pair $X_i, X_j$ to ensure that the estimation error of target agent is lower than $\epsilon$ with the confidence of $1 - \delta$.*

Theorem 1 shows that when adversarial agent uniformly generates random tasks, i.e., when $\zeta$ is close to $\frac{1}{2}$, an infinite number of graph samples are needed for training a better OL agent. The theorem implies that we cannot expect a better performance from the target agent if the adversarial agent is completely random.

### A.2 PROOF OF THEOREM 1

Inspired by the proof process in Yang et al. (2021). For a hypothesis, $\psi \in \Psi$ on target agent, suppose the prediction error of value function is $\epsilon$. We have four cases: (1) the adversarial agent generates an in-distribution graph, and the target agent predicts the right relationship. We have $(1 - \epsilon)(1 - \zeta)$, (2) the adversarial agent generates an in-distribution graph. The target agent predicts the wrong relationship, we have $\epsilon(1 - \zeta)$, (3) the adversarial agent generates an out-of-distribution graph. The target agent predicts the right relationship, we have $(1 - \epsilon)\zeta$ and (4) the adversarial agent produces an out-of-distribution graph, and the target agent predicts the right relationship $\epsilon\zeta$. From the four cases, we know that the actual mismatching from observation data (in-distribution) and target prediction can be concluded as cases (2) and (3). Therefore the mismatching probability equals $(1 - \epsilon)\zeta + \epsilon(1 - \zeta)$. Supposing that the prediction error is larger than $\epsilon$. We have the empirical loss of the $f$ larger than that of the optimal $f^*$ if both of the following two statements hold: (i) the empirical loss of the $f$ is smaller than $\zeta + \frac{\epsilon(1-2\zeta)}{2}$ and (ii) the empirical loss of the $f^*$ is larger than $\zeta + \frac{\epsilon(1-2\zeta)}{2}$. The definition of uniform convergence Shalev-Shwartz & Ben-David (2014) demonstrates below helps us to prove the theorem

**Lemma 1** *Let $\Psi$ be a hypothesis class, then for any $\epsilon \in (0, 1)$ and $\psi \in \Psi$, if the number of training samples is $Nm$, where $N$ is the number of tasks, the following formula holds:*

$$\mathbb{P}(|L(\psi) - \hat{L}(\psi)| > \epsilon) < 2|\Psi| \exp\left(-2Nm\epsilon^2\right)$$

*Where $\hat{L}(\psi)$ is the empirical risk over all the tasks and their samples.*

we focus on the graph (task) size $N$ in each graph from the above lemma. For statement (i), since the prediction error of the target agent is larger than $\epsilon$; the expected loss is larger than $\zeta + \epsilon(1 - 2\zeta)$.

---

[1]The support of $g$ is defined as the smallest closed set containing all points not mapped to zero.

Then, if the empirical loss $L(\psi)$ is smaller than $\zeta + \frac{\epsilon(1-2\zeta)}{2}$, then $|L(\psi) - \hat{L}(\psi)|$ in above lemma is larger than $\frac{\epsilon(1-2\zeta)}{2}$, when the graph (task) size is larger than $\frac{2\log\left(\frac{2|\Psi|}{\delta}\right)}{m\epsilon^2(1-2\zeta)^2}$, we have

$$\mathbb{P}(|L(\psi) - \hat{L}(\psi)| > \frac{\epsilon(1-2\zeta)}{2}) < \delta$$

For statement (ii), we suppose the expectation loss of $\psi^*$ is $\zeta$ (i.e. $L(\psi^*) = \zeta$ ) and the empirical loss $\hat{L}(\psi^*)$ is larger than $\zeta + \frac{\epsilon(1-2\zeta)}{2}$, then $|L(\psi^*) - \hat{L}(\psi^*)|$ in above lemma should be larger than $\frac{\epsilon(1-2\zeta)}{2}$. According to the above lemma, when the graph size is larger than $\frac{2\log\left(\frac{2|\Psi|}{\delta}\right)}{m\epsilon^2(1-2\zeta)^2}$, we have

$$\mathbb{P}(|L(\psi^*) - \hat{L}(\psi^*)| > \frac{\epsilon(1-2\zeta)}{2}) < \delta.$$

As a result, both of the above statements hold with a probability smaller than $\delta$. The proof of the theorem is completed.

## B   NETWORK ARCHITECTURE

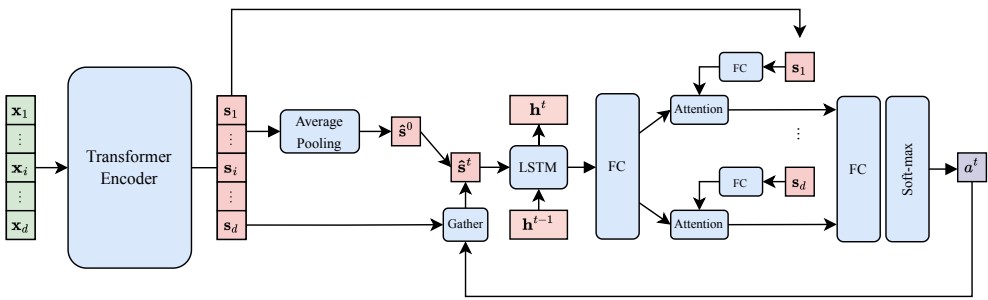

Figure 5: Architecture of the OL agent policy

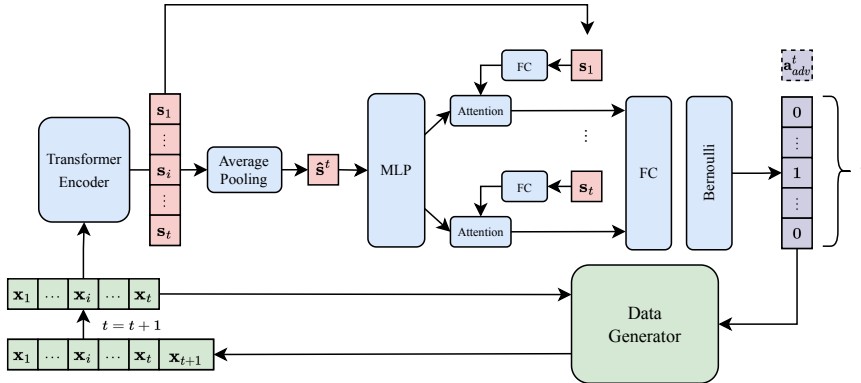

Figure 6: Architecture of the AD agent policy

## C  ALGORITHMS

---

**Algorithm 1** Inferring the order of variables by OL agent

---
**Input:** Observational data $\mathbf{D}$
 1: Initialization: $\Omega = \varnothing$
 2: Draw $N$ samples $[\mathbf{x}_1^t, \ldots, \mathbf{x}_d^t] \in \mathbb{R}^{N \times d}$ from $\mathbf{D}$
 3: $\mathbf{S}_+^0 = \{\mathbf{s}_1, \ldots, \mathbf{s}_d\} \leftarrow \Phi(\{\mathbf{x}_1, \ldots, \mathbf{x}_d\}), \mathbf{S}_-^0 \leftarrow \varnothing$
 4: $\hat{\mathbf{s}}^0 \leftarrow Mean(\mathbf{S}^0)$
 5: **for** $t = 0, \ldots, T - 1$ **do**
 6: $\quad a_t \leftarrow \pi_\phi(\mathbf{S}^t)$
 7: $\quad$ add $\mathbf{s}_{a_t}$ to $\mathbf{S}_-^t$, remove $\mathbf{s}_{a_t}$ in $\mathbf{S}_+^t$
 8: $\quad$ append $a_t$ to $\Omega$
**Output:** $\Omega := (a_0, \ldots, a_{T-1})$

---

**Algorithm 2** DAG Generation by Adversarial Agent

---
**Input:** number of variables $d$
 1: Initialization: $\mathbf{x}_1 := \epsilon_1 \leftarrow \mathcal{N}(0, 1), W_{adv} \leftarrow 0^{d \times d}, T := d$
 2: **for** $t = 1, \ldots, T - 1$ **do**
 3: $\quad \mathbf{S}_{adv}^t = \{\mathbf{s}_1, \ldots, \mathbf{s}_t\} \leftarrow \Phi_{adv}(\{\mathbf{x}_1, \ldots, \mathbf{x}_t\})$
 4: $\quad \mathbf{a}_{adv}^t \leftarrow \pi_\theta(\mathbf{S}_{adv}^t)$
 5: $\quad \epsilon_{t+1} \leftarrow \mathcal{N}(0, 1)$
 6: $\quad$ generate $\mathbf{x}_{t+1}$ with $\mathbf{a}_{adv}^t, \{\mathbf{x}_1, \ldots, \mathbf{x}_t\}, \epsilon_{t+1}$ by Eq. 3
 7: $\quad W_{adv}^{1:t,t} \leftarrow \mathbf{a}_{adv}^t$
 8: $M_{adv} := \{W_{adv}, \{\mathbf{x}_1, \ldots, \mathbf{x}_d\}\}$
**Output:** the adversarial task $M_{adv}$

---

**Algorithm 3** Adversarial Training Framework

---
 1: Initialization: encoder $\Phi$ and $\Phi_{adv}$, OL actor $\pi_\phi$, AD actor $\pi_\theta$, shared critic $V_\psi$ and add pre-sample tasks to training task poll $\mathcal{M}_{train}$
 2: **for** adversarial training epochs **do**
 3: $\quad$ **for** $i$ in OL agent training iterations **do**
 4: $\quad\quad$ sample a batch of training tasks from $\mathcal{M}_{train}$
 5: $\quad\quad$ infer orderings $\Omega$ for each graph with $\Phi$ and $\pi_\phi$ by Alg. 1
 6: $\quad\quad$ evaluate $\Omega$ with corresponding $\mathcal{R}_{OL}(M, \Omega)$
 7: $\quad\quad$ update $\Phi, \pi_\phi$ by Eq. 5
 8: $\quad\quad$ update $V_\psi$ by Eq. 6
 9: $\quad$ **for** $i$ in AD agent training iterations **do**
10: $\quad\quad$ generate tasks $\mathcal{M}_{adv}^i$ with $\Phi_{adv}$ and $\pi_\theta$ by Alg. 2
11: $\quad\quad$ evaluate the reward for generated tasks by OL agent and get the rewards $\mathcal{R}_{OL}(\mathcal{M}_{adv}, \phi)$
12: $\quad\quad$ update $\Phi_{adv}, \pi_\theta$ by Eq. 7
13: $\quad\quad$ update $V_\psi$ by Eq. 6
14: $\quad$ add generated adversarial tasks in last iteration $\mathcal{M}_{adv}$ to $\mathcal{M}_{train}$

---

---

**Algorithm 4** Deployment

---

**Input:** observational data from observational dataset $\mathbf{D}$ with $d$ variables; hyper-parameter: batch size $b$, prune threshold $\epsilon$

1: draw a batch of data samples $\{[\mathbf{x}_1^t, \ldots, \mathbf{x}_d^t] \in \mathbb{R}^{N \times d}\}_{j=1}^b$ from $\mathbf{D}$ with batch size $b$
2: inference a batch of actions $\{\Omega_1, \ldots, \Omega_b\} \leftarrow \pi_\phi(\{[\mathbf{x}_1^t, \ldots, \mathbf{x}_d^t]\}_{j=1}^b)$
3: compute the BIC score of each action $\{s_1^{BIC}, \ldots, s_b^{BIC}\} = \mathcal{S}^{BIC}(\{\Omega_1, \ldots, \Omega_b\})$
4: select the best ordering $\Omega_{max} := \arg\max_i \mathcal{S}^{BIC}(\Omega_i)$
5: generate fully-connected DAG $\mathcal{G}^{full}$ from $\Omega_{max}$
6: prune $\mathcal{G}^{full}$ to get the final adjacency matrix $W^{pruned}$

**Output:** $W^{pruned}$

---

## D  HYPER-PARAMETERS

Table 3: Model parameters for linear and non-linear models

|  | Linear Model | Non-linear Model |
|---|---|---|
| batch size | 64 | 32 |
| encoder heads | 8 | 8 |
| encoder blocks | 3 | 3 |
| encoder dropout rate | 0.1 | 0.1 |
| encoder hidden dim | 1024 | 1024 |
| input dim | 64 | 128 |
| embed dim | 64 | 128 |
| OL actor hidden dim | 64 | 128 |
| AD actor hidden dim | 64 | 128 |
| actor lr | $10^{-4}$ | $10^{-4}$ |
| critic lr | $10^{-3}$ | $10^{-3}$ |

## E  ADDITIONAL EXPERIMENT RESULTS

### E.1  GENERALIZABILITY OF AGCORL

We first evaluate the generalizability of our method in sample cases. We train an OL agent on 10-node Linear Gaussian(LG) tasks and then transfer to larger LG tasks (Table 4) and Linear Non-Gaussian tasks (Table 5). Table 4 shows that even though the task size slightly affects the transfer performance in LG cases, the performance is still satisfying (Normalized SHD (NSHD) increases slightly and TPR drops slightly). Table 5 also shows that the noise type will only slightly affect the transfer performance.

Table 4: Empirical results for Transfer to Different Size

| Task Size | Noise | SHD | TPR | NSHD |
|---|---|---|---|---|
| 10 | Gaussian | $0_{\pm 0.0}$ | $1_{\pm 0.0}$ | $0_{\pm 0.0}$ |
| 20 | Gaussian | $1.6_{\pm 0.7}$ | $0.987_{\pm 0.02}$ | $0.8_{\pm 0.4}$ |
| 30 | Gaussian | $4.2_{\pm 1.9}$ | $0.987_{\pm 0.02}$ | $1.4_{\pm 0.6}$ |
| 40 | Gaussian | $9.1_{\pm 4.4}$ | $0.98_{\pm 0.03}$ | $2.3_{\pm 1.1}$ |
| 50 | Gaussian | $13.8_{\pm 6.8}$ | $0.971_{\pm 0.04}$ | $2.7_{\pm 1.4}$ |

To evaluate the transfer performance of AGCORL to tasks with different function types, we train an agent on GP tasks and transfer to unseen GP, quadratic, and MLP tasks (Table 6). The transfer performance to different function types is not as good as to the same GP but is better than a random policy. Therefore, if we do not have enough clue about the function type, we can use several policies trained on different function types to produce candidate orderings, then select the best based on the BIC score.

Table 5: Empirical results for Transfer to Different Noise

| Task Size | Noise | SHD | TPR |
|-----------|-------|-----|-----|
| 10 | Gaussian | $0_{\pm 0.0}$ | $1_{\pm 0.0}$ |
| 10 | Exp | $0.5_{\pm 0.4}$ | $0.993_{\pm 0.01}$ |
| 10 | Gumbel | $0.5_{\pm 0.4}$ | $0.993_{\pm 0.01}$ |
| 10 | Uniform | $0.5_{\pm 0.4}$ | $0.993_{\pm 0.01}$ |

Table 6: Empirical results for Transfer to Different Function

| Function | 10 nodes | | | |
|----------|----------|----------|----------|----------|
| | 1 edge | | 4 edges | |
| | TPR | SHD | TPR | SHD |
| GP | $0.74_{\pm 0.04}$ | $2.5_{\pm 3.1}$ | $0.36_{\pm 0.06}$ | $13.6_{\pm 8.7}$ |
| Quadratic | $0.56_{\pm 0.10}$ | $4.4_{\pm 1.5}$ | $0.43_{\pm 0.16}$ | $18.0_{\pm 5.4}$ |
| MLP | $0.70_{\pm 0.24}$ | $4.2_{\pm 3.6}$ | $0.55_{\pm 0.16}$ | $16.8_{\pm 5.3}$ |
| Random | $0.45_{\pm 0.17}$ | $5.4_{\pm 3.2}$ | $0.17_{\pm 0.09}$ | $25.3_{\pm 11.6}$ |

## E.2 SCALABILITY OF AGCORL

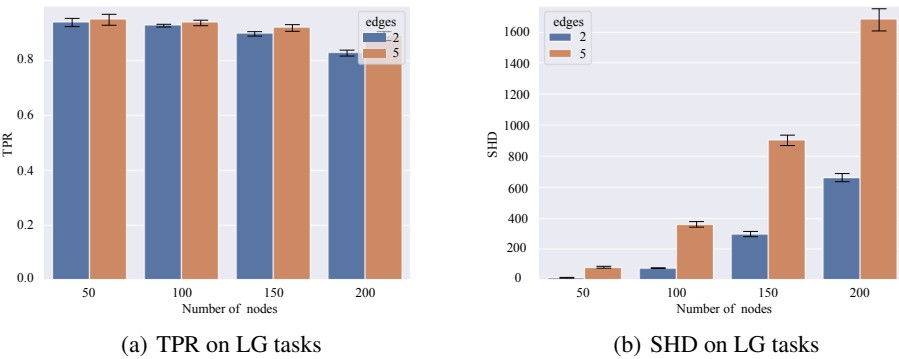

(a) TPR on LG tasks      (b) SHD on LG tasks

Figure 7: AGCORL performance on larger LG tasks

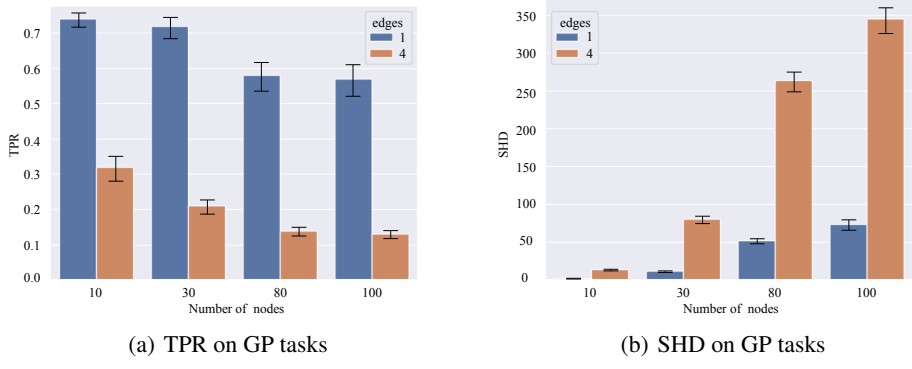

(a) TPR on GP tasks                         (b) SHD on GP tasks

Figure 8: AGCORL performance on larger GP tasks

