# OpenReview forum: "Learning Scalable Causal Discovery Policies with Adversarial Reinforcement Learning"
_ICLR.cc/2024/Conference — Submitted to ICLR 2024_

### Official Review · Reviewer_L2qU · 2023-10-26

**Soundness:** 2 fair
**Presentation:** 2 fair
**Contribution:** 1 poor
**Rating:** 5
**Confidence:** 4

**Summary:**

This paper introduces AGCORL, a new method for learning causal graph structures from observational data. Unlike traditional search-based methods, AGCORL focuses on training reusable causal discovery policies that can generalize to larger tasks efficiently. It uses an OL agent to deduce variable order directly from data and an AD agent to enhance the OL agent's generalizability. The paper shows that AGCORL outperforms existing methods in terms of runtime and solution quality through theoretical and empirical evaluations.

**Strengths:**

- The introduction, related work, and preliminary parts are well organized and clearly conveyed.
- Using an adversarial RL framework to learn the causal graph from observational data seems relatively novel in this field.

**Weaknesses:**

Starting from section 4, the writing of this paper becomes quite chaotic. The authors did not introduce the technical background of how to combine RL and causal discovery, which left me feeling lost. Many symbols are used without prior definitions. I still do not understand what 'tasks' refers to in this paper.

**Questions:**

- The most important aspect of this paper is the introduction of a method using adversarial samples to guide network training. Furthermore, it designs an AD agent to generate a ground-truth graph used as a reward. However, how can we ensure that the tasks generated by this AD agent are helpful to the learning of the OL agent?
- In the paper, a combination of adversarial learning and reinforcement learning is used for training. It's well-known that the convergence of these two methods is a significant challenge. Are there any techniques that can help the convergence of the algorithm?
- The generalizability of DAG learning is a rather werid concept. Why is it that actions can be generalized from a small graph to a larger one? Does this require certain assumptions? Do the causal mechanisms need to be consistent? Can actions learned on linear data be generalized to nonlinear data?
- In the experimental section, the authors only used 500 observations for 50 nodes and 100 nodes. This setting is somewhat challenging. What if we increase the number of observations? Or reduce the number of nodes?
- Why wasn't Notears-MLP compared for nonlinear data? In my experience, this method is actually more robust.
- Do the authors consider this time comparison to be fair? You not only need to find the order but also apply methods like CAM for further pruning. As a DAG learning method, this time also needs to be counted in. Moreover, RL-based methods are indeed quite slow. It would be helpful if the authors could provide some training logs.
- Maximizing the BIC score to learn a causal graph is theoretically grounded and can ensure identifiability. So, how can the method learned using the AD agent ensure that the ground-truth graph is learnable?

---

> ### Author Response · Authors · 2023-11-19
> **Response to Reviewer L2qU (Part 1)**
>
> We thank Reviewer L2qU for your useful feedback. Please see our responses below.
>
> **Q1:** On the readability of the paper.
>
> **A1:** We apologize for the confusion caused. The concept of 'task' was introduced in section 3 under 'Causal Discovery Task & BIC Score.' Actually, Section 3 provides some background of using RL for causal discovery. We would update our paper to provide more clear and comprehensive background information. Please check the new version of the paper later.
>
> **Q2:** The most important aspect of this paper is the introduction of a method using adversarial samples to guide network training. Furthermore, it designs an AD agent to generate a ground-truth graph used as a reward. However, how can we ensure that the tasks generated by this AD agent are helpful to the learning of the OL agent?
>
> **A2:** Referring to Algorithm 3 (Adversarial Training Framework), the OL agent and the AD agent are trained iteratively. The AD agent's reward is correlated with the OL agent's performance on adversarial tasks it generates at the previous interation. Thus, the AD agent is incentivized to generate challenging tasks that the OL agent struggles with. This adversatial training framework helps to improve the robustness of the OL agent so that it can solve unseen and poteitially larger tasks.
>
> Our experimental results in Table 1&2 also support this: AGCORL, with adversarial training, outperformed CORL-P, which is trained using tasks generated from a non-adversarial prior distribution.
>
>
> **Q3:** It's well-known that the convergence of these two methods is a significant challenge. Are there any techniques that can help the convergence of the algorithm?
>
> **A3:** Thanks for raising this good question. Actually, we implemented a task pool strategy to stablize training (see Figure 1). We found that training the OL agent solely on tasks generated by the AD agent in the last epoch caused instability and forgetting of previously learned DAGs. By creating a task pool that stores all tasks and from which the OL agent samples for training, we significantly stabilized the training process. Figure 4-left illustrates that the training process is successfully stablized.
>
> **Q4:** The generalizability of DAG learning is a rather werid concept. Why is it that actions can be generalized from a small graph to a larger one? Does this require certain assumptions? Do the causal mechanisms need to be consistent? Can actions learned on linear data be generalized to nonlinear data?
>
> **A4:** The concept of generalizability in DAG learning parallels research in combinatorial optimization where a generalizable solver could solve larger problems than that used for training. Even in the context of causal discovery, this concept is not new. For example, existing works use **supervised learning** to train reusable networks for solving larger tasks thank training [1,2,3]. But there is no reusable RL based methods for causal discovery. Our work fills this gap by developing a reusable and generalizable RL policy. Specifically, our work focuses on generalizing from tasks with small DAGs to those with larger DAGs, under the assumption that the local patterns of structures and the underlying SCM are shared between small the larger DAGs. This also aligns with the existing works mentioned above. We will describe the concept of generalization more clearly in the paper. Note that the generalization is not guaranteed if the data distributions are largely different (e.g., data generated under linear SCM and non-linear SCM).

---

> > ### Comment · Reviewer_L2qU · 2023-11-23
> > **Generalizability**
> >
> > Thank you for your detailed response, which has clarified certain concerns for me.
> >
> > Despite this clarification, my concern about the generalizability of this method persists. The paper develops a generalizable RL policy on small causal discovery tasks with the goal of scaling up to learn large causal graphs. However, in practical scenarios, we typically lack prior information about the new task. Consequently, the disparity between the learned policy and the actual task remains uncertain. Additionally, the absence of theoretical assurances regarding the recovery of the ground-truth large graph in the presence of varying data distributions raises concerns.
> >
> > While it is reasonable to anticipate that this method could exhibit effective generalization in similar tasks, its practical applicability appears constrained when confronted with novel tasks featuring distribution shifts from the training set. In such cases, the method's utility seems to be limited.

---

> > > ### Author Response · Authors · 2023-11-23
> > > **Response about generalizability**
> > >
> > > Thank you for raising important points about the generalizability of our causal discovery method. Like established methods including LiNGAM [1] and PNL [2], our approach builds on specific assumptions about underlying causal structures. While these assumptions are foundational, they do introduce limitations, particularly when faced with data distributions not aligned with those assumed during development.
> > >
> > > In response to challenges in practical scenarios, where detailed information on new tasks may be limited, our strategy involves training on a broader range of SCM distributions. This approach is detailed in the Deployment section, where we describe the use of multiple Online Learning (OL) agents trained on diverse data distributions. These agents can be applied to unknown tasks, with the best outcomes selected based on the Bayesian Information Criterion (BIC) score.
> > >
> > > This strategy, along with our novel application of adversarial reinforcement learning to train a reusable causal discovery policy, positions our method as a valuable contribution to the field. While acknowledging current limitations, we are optimistic about further research enhancing the method's robustness and adaptability in varied practical situations.
> > >
> > > [1] Shimizu et al., 2006, A linear nonGaussian acyclic model for causal discovery.
> > >
> > > [2] Zhang et al. 2012, On the Identifiability of the Post-Nonlinear Causal Model.

---

> > > > ### Comment · Reviewer_L2qU · 2023-11-23
> > > >
> > > > Thank you for your reply! I'm aware that you utilized the BIC score for agent selection. However, if the top-performing agent struggles to generalize to a new task, your method would fail to find the causal structure for the new task. I recommend incorporating additional related experiments and illustrating this as a limitation of your current method. This would provide a more comprehensive understanding while leaving the theoretical analysis for a future audience capable of handling them.

---

> ### Author Response · Authors · 2023-11-19
> **Response to Reviewer L2qU (Part 2)**
>
> **Q5:**  Why wasn't Notears-MLP compared for nonlinear data? In my experience, this method is actually more robust.
>
> **A5:** We recognize Notears-MLP[4]'s robustness in certain nonlinear contexts. Yet, we compare our AGCORL with a stronger baseline GraN-DAG [5] in the non-lienar experiments and also use Gaussian process with additive noise to generate the data. GraN-DAG has been proven much more effective than Notears-MLP. According to [5], in terms of SHD metric, GraN-DAG scores significantly better than Notears-MLP (1.7 vs. 12.2 in 10ER1 and 8.3 vs. 32.6 in 10ER4 settings). These results align with our findings, therefore we choose GraN-DAG as the more appropriate and robust baseline in our work.
>
> **Q6:** Do the authors consider this time comparison to be fair? You not only need to find the order but also apply methods like CAM for further pruning. As a DAG learning method, this time also needs to be counted in. Moreover, RL-based methods are indeed quite slow. It would be helpful if the authors could provide some training logs.
>
> **A6:** The run times reported in Table 1&2 are the whole time solving for the tesing tasks, including the pruning procedure. Therefore, we believe this is a fair comparison.
>
> Unlike existing RL methods such as CORL, our method learns a reusable causal discovery policy that requires only one forward inference during testing, which significantly saves time for solving multiple testing tasks. By contrast, non-reusable methods like Notears, Grad-DAG, CAM and CORL involve time-consuming procedures like gradient descent, heuristic and reward-guided searches for each testing task.
>
>
> **Q7:** Maximizing the BIC score to learn a causal graph is theoretically grounded and can ensure identifiability. So, how can the method learned using the AD agent ensure that the ground-truth graph is learnable?
>
> **A7:** Maximizing GTR indeed ensures that the true DAG order is identified. Note that GTR directly counts the reversed edges, so if there are wrong oders in inferred DAG, it will definitely be punished in the GTR score of the inferred DAG. Therefore, GTR is an effective and reliable metric that captures the order correctness of a DAG.
>
> Since the adversarial DAGs are generated by the AD agent during training, we know the ground truth structure of these DAGs and therefore could easily compute the GTR to measure the distance of inferred DAGs and adversarial DAGs.
>
> [1] Li et al. (2020). Supervised Whole DAG Causal Discovery.
>
> [2] Lorch et al. (2022). Amortized Inference for Causal Structure Learning.
>
> [3] Ke et al. (2022). Learning to Induce Causal Structure.
>
> [4] Zheng et al. (2019). Learning Sparse Nonparametric DAGs.
>
> [5] Lachapelle et al. (2020). Gradient-Based Neural DAG Learning.

---

> > ### Comment · Reviewer_L2qU · 2023-11-23
> > **Identifiability**
> >
> > The identifiability that I said here is about the model identifiability on the new task since the identifiability is quite important for causal discovery.

---

> > > ### Author Response · Authors · 2023-11-23
> > > **Response about identifiability**
> > >
> > > Thank you for your response. In addressing model identifiability in causal discovery, it's crucial to first define a specific model and its parameter space. Our method, akin to several other search-based approaches, operates without fitting an explicit model to the data. Therefore, traditional discussions of model identifiability, which typically pertain to predefined models and their parameters, are not directly applicable to our methodology. Our focus is on the search process rather than model fitting, placing our method outside the usual scope of model identifiability analysis.

---

> > > > ### Comment · Reviewer_L2qU · 2023-11-23
> > > >
> > > > Ensuring model identifiability entails two key aspects: (1) under appropriate assumptions on data distribution, the model can be uniquely determined, either in its true form or within a defined equivalence class; (2) **your proposed method has the capability to accomplish this identification.** My concern is that how your method can ensure to identify the model.

---

> ### Comment · Reviewer_L2qU · 2023-11-23
>
> Thanks for your positive response! I decided to increase my score from 3 to 5.

---

### Official Review · Reviewer_GfFV · 2023-10-31

**Soundness:** 3 good
**Presentation:** 2 fair
**Contribution:** 2 fair
**Rating:** 5
**Confidence:** 3

**Summary:**

The authors propose AGCORL, an RL based approach to directly output the topological order from an input dataset. The adversarial component generates graphs that are difficult for the ordering agent to learn. The training data is generated from a particular set of SCMs (e.g., linear or GP models).

**Strengths:**

The paper proposes a RL formulation of learning the topological order from an input observational dataset. At each step, the agent takes an action of taking a variable and adding it to the ordered set. The adversarial agent attempts to generate graphs such that the ordering agent achieves low reward. The authors in the experiments show that the agent generalizes to larger graph distributions than in training. They also show that on a real-world protein dataset, they achieve the lowest SHD, showing some promising generalization beyond the training distribution.

**Weaknesses:**

### Clarity

Overall, I found the paper easy to read and follow. However, I think the writing in certain sections can be improved. For example, the related work does not explain the prior works CORL in enough detail. It would be good if the text made more efforts to explicitly distinguish their work from CORL. It is not currently clear to me what precisely the innovation is relative to CORL. Moreover, it would also make it easier to interpret why AGCORL and CORL have such different runtimes.


### Experiments

One weakness in the experiments is that the authors only attempt to test generalization to larger graph instances. However, one limitation of AGCORL is that the training data itself is synthetic (so limited to linear gaussain or GP SCMs). The authors only test on the same SCM distributions. It is not clear what would happen if a different distribution of SCMs arrives at test time (e.g., what if the test SCM used a different GP kernel than during training). In these cases, the baselines might do better. Or what if the graph distribution at test time is not ER or scale-free? More experiments understanding generalization would be useful.

Re metrics:
Since AGCORL only learns the topological ordering, why don't the authors use metrics that test the how good the ordering is relative to the true ordering? Using variable selection on top of the learned ordering seems to make the results harder to interpret as it is not clear whether the errors are due to the wrong ordering or wrong variable selection.

Re contribution of the adversarial component:
In Fig. 4 left, the authors show the impact of the adverarail training. However, the curves in Fig 4 do not in themselves suggest that adversarial training helps. What would be useful is training AGCORL with and without the adversarial parts and comparing the metrics. What we care about is whether the metrics at test time graphs improve or not.

Type in section 5.2: Table 3 -> Table 2.

**Questions:**

Re reinforcement learning formulation:
Can the authors comment on why they chose to formulate this as an RL problem as opposed to a supervised learning task? The state is a tuple of $ < s^{+}, s^{-} >$ which means that for a given action $a$, the next state can be deterministically computed. This is quite different from a standard RL setup where the states are stochastically generated by the environment using $p(s_{t+1}|s_t, a_t)$. Moreover, you only get a reward after the entire episode ends (i.e., you have the full ordering). So it seems like you could also just treat this is a supervised learning problem where you directly output an ordering. The challenge is probably computing a differentiable loss between the true ordering and predicted ordering. I think many existing losses exist in the literature for this task. Can the authors comment on this? It seems like directly using supervised formulations might be more efficient (since this RL formulation is essentially like zeroth-order optimization to learn the ordering).

Re runtime:
In Table 2, why does AGCORL take 11m to run for a 30-node DAG? Whereas in Table 1, it takes 19.2s for a 100-node graph. At inference time, my understanding is that you take the trained policy agent and just take d-steps to compute the ordering. So why is it slower for a 30-node DAG?

---

> ### Author Response · Authors · 2023-11-19
> **Response to Reviewer GfFV**
>
> We thank the reviewer for your constructive feedback. Please see our responses below.
>
> **Q1:** Comparing with CORL.
>
> **A1:** While CORL also uses RL for causal discovery tasks, it is not scalable by design. For each given testing task, CORL will run a RL training process to find the optimal causal graph. By contrast, AGCORL uses adversarial tasks generated by the AD agent to learn a policy, which could directly output solutions for a given testing task (possibly larger than the training ones). Thus, since the AGCORL policy is reusable and scalable, it significantly reduces testing time compared to CORL. We will revise the related work section to more explicitly highlight these distinctions and their impact on runtime differences.
>
> **Q2:** On the transferability.
>
> **A2:** We thank the reviewer for raising this valuable question.
> Actually, experimental results in Table 6 and Section 5.3 show that AGCORL achieves comparable transferability compared to baselines. However, note that AGCORL focuses on **scalability** (generalization from smaller to larger graphs) and most experiments are designed to justify this point. In addition, it is a common practice in causal discovery to pre-assume some specific SCM types and learn from synthetic data, especially when we do not have any prior knowledge on the testing tasks. However, we aggree with the reviewer that transferability is also an important question to explore. We will leave it for future work.
>
> **Q3:** Why do we use variable selection on the top of the algorithm, instead of directly test how good the ordering is relative to the true ordering?
>
> **A3:** Actually, the mapping from a DAG to its ordering is neither injective nor surjective, meaning multiple correct orderings can exist for a given DAG. Therefore, it is hard to design a metric to directly evaluate the inferred orderings. Therefore, followed by existing works, we first construct fully-connected DAG using the infered orderings and then use standard pruning algorithms to get the final DAG. In practice, the pruning algorithms can work efficiently with only minor mistakes, since the solution space of DAG has been substantially reduced by the infered orderings. By applying a consistent pruning algorithm across all testing methods, the comparision is more fair and reliable.
>
> **Q4:** On the contribution of the adversarial component: In Fig. 4 left, the authors show the impact of the adverarail training. However, the curves in Fig 4 do not in themselves suggest that adversarial training helps. What would be useful is training AGCORL with and without the adversarial parts and comparing the metrics. What we care about is whether the metrics at test time graphs improve or not.
>
> **A4:** Fig 4 illustrates how adversatial training (AD agent) helps to improve the training of the OL agent. The experiments on AGCORL with and without the adversarial parts, as the reviewer mentioned, is shown in Table 1 and Table 2, where the CORL-P is actually AGCORL without adversarial part. To train CORL-P, we sample training tasks from a prior distribution without adversary. The empirical results show that the CORL-P performs worse than AGCORL, which demonstrates the contribution of the adversarial component. We will rename CORL-P as AGCORL-without-AD to avoid the confusion caused. Thanks for pointing out this.
>
> **Q5:** Why do we use reinforcement learning?
>
> **A5:** Compared with supervised learning based approaches which directly output a DAG solution for a given causal discovery task, RL based methods decompose the solution construction as a sequential decision making process. The decomposition of the task lowers the difficulty of the problem and potentially leads to higher quality of solutions. In addition, deterministic transition and episodic reward are very common in RL environments such as robotics. So this does not weaken the motivation of using RL.
>
> **Q6:** On the runtime: In Table 2, why does AGCORL take 11m to run for a 30-node DAG? Whereas in Table 1, it takes 19.2s for a 100-node graph. Why is it slower for a 30-node DAG?
>
> **A6:** The runtime difference between a 30-node DAG and a 100-node graph in AGCORL is primarily due to the nature of the data being processed. For nonlinear data, as in the 30-node case, the pruning process is more time-intensive compared to linear data, which is the case for the 100-node graph. Since AGCORL, along with CAM and CORL, first determines the node order and then applies a variable selection algorithm for pruning, the complexity of the data type significantly impacts the overall runtime. Hence, the 30-node nonlinear task requires more time than the 100-node linear task.

---

> > ### Comment · Reviewer_GfFV · 2023-11-22
> > **Response**
> >
> > I thank the authors for their response.
> >
> > Re response to transferability:
> > I agree that showing generalizability to larger graphs is valuable, it is only a single axis of generalization. The difference relative to nonparametric causal discovery (like the PC algorithm) is that when you only train on synthetic data, you are (probably) limited to generalizing to the same set of SCMs. So, for example, even if you train on linear SCMs, what would happen if you encounter a linear SCM with different parameters than the one seen in training? It is not clear that the proposed method would work. So it would have been interesting to see more experiments testing various failure modes of this approach.
> >
> > Re "Actually, the mapping from a DAG to its ordering is neither injective nor surjective, meaning multiple correct orderings can exist for a given DAG."
> > I agree that you would have to somehow account for ordering across the MEC. However, it seems like there are many ways to fix this. Quite simply, you could have reported an accuracy metric that denotes whether the ordering is valid for the true DAG or not.
> > Equivalently, you could have used an oracle pruner (that prunes using the true d-separations). There are also many metrics that evaluate an estimated ordering against a true ordering (including partial ordering as determined via the MEC). I still think the variable selection makes the results harder to interpret. As you mention in a future response, for nonlinear data, pruning takes a long time and it is not obvious that in these cases the errors are dominated by the wrong ordering.

---

> > > ### Author Response · Authors · 2023-11-23
> > > **Response**
> > >
> > > Re response to transferability:
> > >
> > > Thank you for acknowledging the value of our work in demonstrating generalizability to larger graph structures. We recognize that this represents only one dimension of generalization. Indeed, training exclusively on synthetic data, as in our method, may inherently limit generalization to a specific set of Structural Causal Models (SCMs). This raises valid concerns, such as how our method would perform when encountering linear SCMs with parameter variations not seen during training.
> > >
> > > While it's true that encountering significantly different SCM parameters could impact performance, we believe this challenge can be mitigated by allocating more computational resources to train the agent across a broader spectrum of SCM distributions. Such an approach could enhance the method's adaptability to diverse scenarios. Thus, while our current focus is on graph structure generalizability, future iterations of our method, with adequate computational support, hold the potential to address a wider range of generalization challenges, including those highlighted in your query.
> > >
> > > Re metrics and pruning:
> > >
> > > Thank you for your suggestions regarding the evaluation metrics for DAG ordering. However, our primary objective is the accurate identification of DAGs rather than their specific orderings. This focus aligns with the majority of prior work in the field, which typically emphasizes DAG accuracy over ordering precision. Besides, there exist non-ordering-based methods in baselines.  Consequently, we have chosen to prioritize metrics that assess the correctness of the DAGs themselves, in line with these established methodologies.
> > >
> > > Regarding nonlinear data, we acknowledge the increased difficulty in order inference. Despite these challenges, our method has demonstrated superior performance in comparison to all baselines in handling such data. This suggests that, while the complexity of the task increases with nonlinear data, our approach remains effective in accurately identifying the underlying DAG structures.

---

### Official Review · Reviewer_Wgj3 · 2023-11-01

**Soundness:** 2 fair
**Presentation:** 3 good
**Contribution:** 2 fair
**Rating:** 5
**Confidence:** 3

**Summary:**

The paper proposes an adversarial reinforcement learning framework for causal structure learning named AGCORL.
AGCORL mutually trains an order learning agent (OL agent) and an adversarial agent (AD agent).
The OL agent learns a general policy that infers the causal order of variables, while the AD agent generates challenging tasks for the OL agent.
The trained OL agent avoids calculating the computationally expensive Bayesian Information Criterion (BIC score), making it suitable for large-scale tasks.
Experiments show that the trained OL agent can generalize to new tasks that are different from the tasks used in training.

**Strengths:**

- The proposed Ground Truth Reward (GTR) enables AGCORL to efficiently train both agents without calculating the time-consuming BIC score.

- The OL agent shows good scalability ($t$ in Table 2).
Additionally, an OL agent trained on one task can transfer to other tasks.
In particular, it maintains good performance when transferred to tasks with different noise (Table 5).

**Weaknesses:**

- The OL agent is trained on synthetic data generated by an explicit Structural Causal Model (SCM).
Therefore, the OL agent may not be suitable for real-world data tasks where we do not know the mathematical relationship between the variables.
In such cases, the authors suggest to use multiple OL agents trained with different function types to infer candidate orderings, and then calculate their BIC score to select the best one (Appendix E).
However, this approach could compromise the fast running time, which is a main advantage of the proposed work.

- The paper lacks theoretical and/or qualitative explanations that compare the BIC score and the newly proposed GTR.
It seems important to clarify whether GTR can always be a surrogate for BIC.
If so, AGCORL could replace frameworks that calculate the BIC score, even for small-scale tasks.
If not, in certain circumstances, it will still be necessary to use BIC score based algorithms, even for large-scale tasks.

- Theorem 1 in Appendix A only shows that a random adversarial agent is the worst for training a generalizable order learning agent.
It does not provide any indication of how good the authors' proposed AD agent is.
For example, a theoretical analysis revealing how far the proposed AD agent's $\zeta$ is from 0.5, with some probability, would be helpful.

**Questions:**

- Is there any measure other than the Structural Hamming Distance (SHD) that can distinguish between missing edges and reversed edges?
The authors argue that the OL agent is generalizable to large tasks based on the observation that the True Positive Rate (TPR) only decreases slightly (Table 4).
However, in the same table, the SHD increases.
If the SHD increases because of a large number of reversed edges, it is difficult to conclude that the OL agent generalizes to large tasks.

- Does AGCORL outperform previous BIC score-based algorithms on small-scale tasks too?
If so, can we safely say that AGCORL replaces previous BIC score-based algorithms in all cases?

- For the same $d$ and $\theta$, how does the time complexity of computing the BIC score differ from that of Algorithm 1?

---

> ### Author Response · Authors · 2023-11-19
> **Response to Reviewer Wgj3**
>
> We thank Reviewer Wgj3 for your constructive feedback.
>
> **Q1:** Using multiple OL agents compromise the fast running time, which is a main advantage of the proposed work.
>
> **A1:** It is common in literature to assume some specific SCMs and generalize to real-world data, since the variable relationships are usually unknown. Actually, using multiple OL agents to infer candidate orderings is a practical approach to adapt to different real-world data types. While this may increase the computational load, the time complexity for inference remains efficient at $O(ld^2)$, where $l$ is the number of OL agents and $d$ is the task size. This is notably more efficient than other methods with complexities of $\Omega(d^3)$, especially for large $d$. Thus, our approach maintains a balance between adaptability to real-world data and the advantage of fast runtime, which is a key feature of our method.
>
> **Q2:** Whether GTR can always be a surrogate for BIC.
>
> **A2:** The key difference between GTR and BIC is that the computation of GTR requires ground-truth DAG, which is generated by the AD agent. Of course, in small scale tasks we can also apply AGCORL to achieve competitive performance (see Table 1&2). However, the necessity of appying AGCORL in small-scale tasks is not that significant, because AGCORL is designed for improving scalability.
>
> **Q3:** Theorem 1 in Appendix A only shows that a random adversarial agent is the worst for training a generalizable order learning agent. It does not provide any indication of how good the authors' proposed AD agent is. For example, a theoretical analysis revealing how far the proposed AD agent's $\zeta$ is from 0.5, with some probability, would be helpful.
>
> **A3:** Since the AD agent's $\zeta$ is closely related to experimental settings such as the number of nodes and SCMs, we can hardly estimate $\zeta$ accurately. However, Theorem 1 reveals that improving $\zeta$ by the AD agent could significantly reduce the number of required training tasks, hence successfully justifies the necessity of using the AD agent.
>
> **Q4:** Is there any measure other than the Structural Hamming Distance (SHD) that can distinguish between missing edges and reversed edges? The authors argue that the OL agent is generalizable to large tasks based on the observation that the True Positive Rate (TPR) only decreases slightly (Table 4). However, in the same table, the SHD increases. If the SHD increases because of a large number of reversed edges, it is difficult to conclude that the OL agent generalizes to large tasks.
>
> **A4:** The increase in SHD with larger graph sizes and more edges is a typical occurrence in causal discovery. To provide a more nuanced perspective, considering normalized SHD (SHD divided by the number of edges) offers a clearer view. This normalized metric exhibits only a slight increase on larger tasks, which is consistent with the changes observed in the True Positive Rate (TPR). This alignment suggests that the increase in SHD does not predominantly stem from a large number of reversed edges, supporting the generalizability of the OL agent to larger tasks.
>
> **Q5:** Does AGCORL outperform previous BIC score-based algorithms on small-scale tasks too? If so, can we safely say that AGCORL replaces previous BIC score-based algorithms in all cases?
>
> **A5:** In our experiments, AGCORL achieves competitive performance to existing BIC score-based algorithms on small-scale tasks (see Table 1&2). However, since the AGCORL is designed to improve scalability, we cannot guarantee that AGCORL outperforms baselines in all small-scale tasks. Therefore, it is more safe to try both algorithms in small-scale tasks and use AGCORL in large-scale tasks where BIC-based algorithms are not applicable.
>
> **Q6:** For the same $d$ and $\theta$, how does the time complexity of computing the BIC score differ from that of Algorithm 1?
>
> **A6:** In the linear case, computing the BIC score for one node with $n$ parents is $O(n^3)$, making the overall complexity for an order $O(d^4)$. By contrast, Algorithm 1 uses a Transformer model, which has a time complexity of $O(t^2)$ at step $t$, resulting in a total complexity of $O(d^3)$.

---

> > ### Comment · Reviewer_Wgj3 · 2023-11-23
> > **Thanks**
> >
> > Thank you for your responses. I will take these into account in the final evaluation.

---

### Official Review · Reviewer_yLLZ · 2023-11-07

**Soundness:** 3 good
**Presentation:** 3 good
**Contribution:** 3 good
**Rating:** 6
**Confidence:** 4

**Summary:**

The paper proposes an adversarial reinforcement learning framework for efficient causal discovery on observational data. The framework consists of two agents, OL (ordering learning) and AD (Adversarial agent), respectively, in a zero-sum setting in which AD is adding the tasks that OL has most room to improve in performance. Authors show both theoretically and empirically that the adversarial agent helps for better generalisability with higher data efficiency, and their approach is comparable or better to the previous approaches.

**Strengths:**

-The paper is very well written.  So clarity is in general very good.

- This is a well-defined and a very important area of AI research, hence it is very relevant.

-The approach is interesting and certainly original, and the results are in general promising. Significance is non-trivial although limited (see weaknesses).

**Weaknesses:**

-Code is not shared so the results are irreproducible.

- So many nodes but so little edges. E.g. Only 2 and 5 Table 1. and 1 to 4 with Table 2. This makes me skeptical about the relevance of the results. (See questions)

-   the construction of adversarial graph is unclear to me.  Caption of Figure 3 attempts to explain how does it work, but it is not clear to me still. It should be improved. (See the question)  In general would be great to write down the general procedure. (if it's already there, apologies).


-It would be good give high level intuition on how the employed pruning algorithm works (especially at the end of the deployment subsection). Currently, just referring to the appendix without sharing the idea obscures it.



Minor issues:

-  typo at conclusion : AL -> AD

**Questions:**

-Maybe a silly question but  could you help me understand  Figure 3: how does the action sets and added nodes and the edge works?  (Please also revise the text to make it more clear.)

-So many nodes but so little edges. E.g. Only 2 and 5 Table 1. and 1 to 4 with Table 2. I wonder how does your results are when  it comes to different sparsity, node vs edge is more balanced, or dense?

---

> ### Author Response · Authors · 2023-11-17
> **Responses to Reviewer yLLZ**
>
> We thank Reviewer yLLZ for your helpful feedback. Please see our responses below.
>
> **Q1:** Code is not shared so the results are irreproducible.
>
> **A1:** We promise to share our code upon acceptance of this paper.
>
> **Q2:** So many nodes but so little edges. E.g. Only 2 and 5 Table 1. and 1 to 4 with Table 2.
>
> **A2:** The term 'edge' in our experiment refers to the average degree of a node in a DAG, not the total edge count. So a 50-node-2-edge graph will have 50*2/2=50 edges in average. We will clarify this by replacing 'edge' with 'degree' in the paper.
>
> **Q3:** The construction of adversarial graph is unclear.
>
> **A3:** In our approach, the construction of adversarial graphs is based on the causal principle that the data of a child node is generated from its parent nodes according to Structural Causal Models (SCMs). The AD agent creates these DAGs where $\mathbf{a}^i_{adv}$ specifies the parent nodes of the $i+1$-th node. For instance, in Fig 3, $\mathbf{a}^3_{adv} = {0,1,0}$ indicates that $X_4$ has $X_2$ as its sole parent node. Consequently, the data for $X_4$ is generated using the SCM: $X_4 = f(X_2) + \epsilon_4$. We apologize for any confusion caused by a labeling error in Fig 3, where $X_4$ was incorrectly marked as $X_2$. We will correct this in the figure caption to enhance clarity. This general procedure of adversarial graph construction underpins our methodology and aligns with established causal principles.
>
> **Q4:** It would be good give high level intuition on how the employed pruning algorithm works.
>
> **A4:** Our pruning algorithm, in essence, constructs a fully-connected DAG and then selectively removes edges to form the final DAG. This process involves evaluating the edge significance (e.g., using regression techniques), with a threshold to determine which edges to be pruned. For linear cases, we simple apply linear regression and a selected threshold (0.3 in our experiments). For nonlinear cases, we use a more complex pruning method outlined in CAM [1]. Since the pruning algorithms are not the focus of this work, we did not elaborate them in our paper. But we will add high-level descriptions of them in the deployment part. Thanks for your advice!
>
> [1] Peter Bühlmann, Jonas Peters, and Jan Ernest. CAM: Causal additive models, high-dimensional order search and penalized regression. The Annals of Statistics, 42(6):2526–2556, 2014.

---

### Author Response · Authors · 2023-11-19
**Response to Reviewer Comments: Detailed Revision Overview**

We have carefully reviewed and addressed the comments from the reviewers, making several revisions to enhance the clarity and accuracy of our paper. The changes are highlighted in **red** for easy identification:

1. Corrections of typos: a) $X_4$ in Fig 3, b) Table 2 in Section 5.2, and c) AD agent in the conclusion.
2. Expanded description in the Related Works section to clarify our innovations compared to CORL.
3. Updated the last subsection in the Preliminary section to 'Ordering-based Causal Discovery,' offering a clearer depiction of CORL.
4. Enhanced details in Fig 3's caption for improved clarity.
5. Introduced a **NSHD** (Normalized Structural Hamming Distance) column in Table 4, substantiating our assertion that generalizability only marginally decreases with larger tasks.

---

### Author Response · Authors · 2023-11-22
**Have our responses resolved your concerns?**

Dear reviewers:

Thank you very much for your insightful reviews and constructive suggestions for our work. We have posted our response and tried our best to resolve your questions and doubts. Have our responses resolved your questions and doubts? Your feedback is significant to us.

Thank you again for your time and patience!

Sincerely,
The authors.

---

### Meta-Review · Area_Chair_uh9R · 2023-12-09

**Metareview:**

The paper introduces AGCORL, an adversarial reinforcement learning framework for causal structure learning. The framework involves two agents, OL (ordering learning) and AD (adversarial), working in a zero-sum setting. The OL agent deduces variable order from data, and the AD agent generates challenging tasks. The paper claims better generalizability and higher data efficiency, comparing favorably to previous approaches.

The strength lies in that 1) The approach is somewhat original and promising. 2) Experimental results show promise, especially on real-world protein data.

The weakness lies in that 1) Lack of clarity in some sections, and writing can be improved. 2) Generalizability concerns, especially with novel tasks featuring distribution shifts.

The reviewers highlight the paper's originality and relevance in proposing an adversarial reinforcement learning framework for causal structure learning. However, concerns are raised about clarity, reproducibility, and generalizability. The rating of the majority reviewers is consistently marginally below the acceptance threshold, which leads to the rejection of this paper.

**Justification For Why Not Higher Score:**

The reviewers highlight the paper's originality and relevance in proposing an adversarial reinforcement learning framework for causal structure learning. However, concerns are raised about clarity, reproducibility, and generalizability. The rating of the majority reviewers is consistently marginally below the acceptance threshold, which leads to the rejection of this paper.

**Justification For Why Not Lower Score:**

N/A

---

### Decision · Program_Chairs · 2024-01-16

Reject